# Minimax statistical learning
# with Wasserstein distances

**Jaeho Lee**     **Maxim Raginsky**
{jlee620, maxim}@illinois.edu[*]

## Abstract

As opposed to standard empirical risk minimization (ERM), *distributionally robust optimization* aims to minimize the worst-case risk over a larger ambiguity set containing the original empirical distribution of the training data. In this work, we describe a minimax framework for statistical learning with ambiguity sets given by balls in Wasserstein space. In particular, we prove generalization bounds that involve the covering number properties of the original ERM problem. As an illustrative example, we provide generalization guarantees for transport-based domain adaptation problems where the Wasserstein distance between the source and target domain distributions can be reliably estimated from unlabeled samples.

## 1   Introduction

In the traditional paradigm of statistical learning [20], we have a class $\mathcal{P}$ of probability measures on a measurable *instance space* $\mathcal{Z}$ and a class $\mathcal{F}$ of measurable functions $f : \mathcal{Z} \to \mathbb{R}_+$. Each $f \in \mathcal{F}$ quantifies the loss of some decision rule or a hypothesis applied to instances $z \in \mathcal{Z}$, so, with a slight abuse of terminology, we will refer to $\mathcal{F}$ as the *hypothesis space*. The (expected) *risk* of a hypothesis $f$ on instances generated according to $P$ is given by

$$R(P, f) := \mathbf{E}_P[f(Z)] = \int_{\mathcal{Z}} f(z) P(\mathrm{d}z).$$

Given an $n$-tuple $Z_1, \ldots, Z_n$ of i.i.d. training examples drawn from an unknown $P \in \mathcal{P}$, the objective is to find a hypothesis $\widehat{f} \in \mathcal{F}$ whose risk $R(P, \widehat{f})$ is close to the minimum risk

$$R^*(P, \mathcal{F}) := \inf_{f \in \mathcal{F}} R(P, f) \tag{1}$$

with high probability. Under suitable regularity assumptions, this objective can be accomplished via Empirical Risk Minimization (ERM) [20, 13]:

$$R(P_n, f) = \frac{1}{n} \sum_{i=1}^{n} f(Z_i) \longrightarrow \min, \ f \in \mathcal{F} \tag{2}$$

where $P_n := \frac{1}{n} \sum_{i=1}^{n} \delta_{Z_i}$ is the empirical distribution of the training examples.

Recently, however, an alternative viewpoint has emerged, inspired by ideas from robust statistics and robust stochastic optimization. In this *distributionally robust framework*, instead of solving the ERM problem (2), one aims to solve the minimax problem

$$\sup_{Q \in \mathcal{A}(P_n)} R(Q, f) \longrightarrow \min, \ f \in \mathcal{F} \tag{3}$$

---

[*]Department of Electrical and Computer Engineering and Coordinated Science Laboratory, University of Illinois, Urbana, IL 61801, USA. This work was supported in part by NSF grant nos. CIF-1527 388 and CIF-1302438, and in part by the NSF CAREER award 1254041.

where $\mathcal{A}(P_n)$ is an *ambiguity set* containing the empirical distribution $P_n$ and, possibly, the unknown probability law $P$ either with high probability or almost surely. The ambiguity sets serve as a mechanism for compensating for the uncertainty about $P$ that inherently arises due to having only a finite number of samples to work with, and can be constructed in a variety of ways, e.g. via moment constraints [9], $f$-divergence balls [8], and Wasserstein balls [16, 11, 5]. However, with the exception of the recent work by Farnia and Tse [9], the minimizer of (3) is still evaluated under the standard statistical risk minimization paradigm.

In this work, we instead study the scheme where the statistical risk minimization criterion (1) is replaced with the *local minimax risk*

$$\inf_{f \in \mathcal{F}} \sup_{Q \in \mathcal{A}(P)} R(Q, f)$$

at $P$, where the ambiguity set $\mathcal{A}(P)$ is taken to be a Wasserstein ball centered at $P$. As we will argue below, this change of perspective is natural when there is a possibility of *domain drift*, i.e., when the learned hypothesis is evaluated on a distribution $Q$ which may be different from the distribution $P$ that was used to generate the training data.

The rest of this paper is organized as follows: In Section 2, we formally present the notion of local minimax risk and discuss its relationship to the statistical risk, which allows us to assess the performance of minimax-optimal hypothesis in specific domains. We also provide an example to illustrate the role of ambiguity sets in rejecting nonrobust hypotheses.

In Section 3, we show that the hypothesis learned with the Empirical Risk Minimization (ERM) procedure based on the local minimax risk closely achieves the optimal local minimax risk. In particular, we provide a data-dependent bound on the generalization error, which behaves like the bound for ordinary ERM in the no-ambiguity regime (Theorem 1), and excess risk bounds under uniform smoothness assumptions on $\mathcal{F}$ (Theorem 2) and a less restrictive assumption that $\mathcal{F}$ contains at least one smooth hypothesis (Theorem 3).

In Section 4, we provide an alternative perspective on *domain adaptation* based on the minimax statistical learning under the framework of Courty et al. [6], where the domain drift is due to an unknown transformation of the feature space that preserves the conditional distribution of the labels given the features. Completely bypassing the estimation of the transport map, we provide a proper excess risk bound that compares the risk of the learned hypothesis to the minimal risk achievable within the given hypothesis class on the target domain (Theorem 4). To the best of our knowledge, all existing theoretical results on domain adaptation are stated in terms of the discrepancy between the best hypotheses on the source and on the target domains.

All proofs are deferred to the appendix.

## 2 Local minimax risk with Wasserstein ambiguity sets

We assume that the instance space $\mathcal{Z}$ is a Polish space (i.e., a complete separable metric space) with metric $d_{\mathcal{Z}}$. We denote by $\mathcal{P}(\mathcal{Z})$ the space of all Borel probability measures on $\mathcal{Z}$, and by $\mathcal{P}_p(\mathcal{Z})$ with $p \geq 1$ the space of all $P \in \mathcal{P}(\mathcal{Z})$ with finite $p$th moments. The metric structure of $\mathcal{Z}$ can be used to define a family of metrics on the spaces $\mathcal{P}_p(\mathcal{Z})$ [21]:

**Definition 1.** *For $p \geq 1$, the $p$-Wasserstein distance between $P, Q \in \mathcal{P}_p(\mathcal{Z})$ is*

$$W_p(P, Q) := \inf_{\substack{M(\cdot \times \mathcal{Z}) = P \\ M(\mathcal{Z} \times \cdot) = Q}} \left( \mathbf{E}_M(d_{\mathcal{Z}}^p(Z, Z')) \right)^{1/p}, \tag{4}$$

*where the infimum is taken over all* couplings *of $P$ and $Q$, i.e. probability measures $M$ on the product space $\mathcal{Z} \times \mathcal{Z}$ with the given marginals $P$ and $Q$.*

**Remark 1.** Wasserstein distances arise in the problem of *optimal transport*: for any coupling $M$ of $P$ and $Q$, the conditional distribution $M_{Z'|Z}$ can be viewed as a randomized policy for 'transporting' a unit quantity of some material from a random location $Z \sim P$ to another location $Z'$, while satisfying the marginal constraint $Z' \sim Q$. If the cost of transporting a unit of material from $z \in \mathcal{Z}$ to $z' \in \mathcal{Z}$ is given by $d_{\mathcal{Z}}^p(z, z')$, then $W_p^p(P, Q)$ is the minimum expected tranport cost.

We now consider a learning problem $(\mathcal{P}, \mathcal{F})$ with $\mathcal{P} = \mathcal{P}_p(\mathcal{Z})$ for some $p \geq 1$. Following [16, 17, 11], we let the ambiguity set $\mathcal{A}(P)$ be the $p$-Wasserstein ball of radius $\varrho \geq 0$ centered at $P$:

$$\mathcal{A}(P) = B_{\varrho,p}^W(P) := \left\{ Q \in \mathcal{P}_p(Z) : W_p(P, Q) \leq \varrho \right\},$$

where the radius $\varrho > 0$ is a tunable parameter. We then define the *local worst-case risk* of $f$ at $P$,

$$R_{\varrho,p}(P, f) := \sup_{Q \in B_{\varrho,p}^W(P)} R(Q, f),$$

and the *local minimax risk* at $P$:

$$R_{\varrho,p}^*(P, \mathcal{F}) := \inf_{f \in \mathcal{F}} R_{\varrho,p}(P, f).$$

## 2.1 Local worst-case risk vs. statistical risk

We give a couple of inequalities relating the local worst-case (or local minimax) risks and the usual statistical risks, which will be useful in Section 4. The first one is a simple consequence of the Kantorovich duality theorem from the theory of optimal transport [21]:

**Proposition 1.** *Suppose that $f$ is $L$-Lipschitz, i.e., $|f(z) - f(z')| \leq L d_{\mathcal{Z}}(z, z')$ for all $z, z' \in \mathcal{Z}$. Then, for any $Q \in B_{\varrho,p}^W(P)$,*

$$R(Q, f) \leq R_{\varrho,p}(P, f) \leq R(Q, f) + 2L\varrho.$$

As an example, consider the problem of binary classification with hinge loss: $\mathcal{Z} = \mathcal{X} \times \mathcal{Y}$, where $\mathcal{X}$ is an arbitrary feature space, $\mathcal{Y} = \{-1, +1\}$, and the hypothesis space $\mathcal{F}$ consists of all functions of the form $f(z) = f(x, y) = \max\{0, 1 - y f_0(x)\}$, where $f_0 : \mathcal{X} \to \mathbb{R}$ is a candidate predictor. Then, since the function $u \mapsto \max\{0, 1 - u\}$ is Lipschitz-continuous with constant 1, we can write

$$|f(x, y) - f(x', y')| \leq |y f_0(x) - y' f_0(x')| \leq 2\|f_0\|_{\mathcal{X}} \mathbf{1}\{y \neq y'\} + |f_0(x) - f_0(x')|,$$

where $\|f_0\|_{\mathcal{X}} := \sup_{x \in \mathcal{X}} |f_0(x)|$. If $\|f_0\|_{\mathcal{X}} < \infty$ and if $f_0$ is $L_0$-Lipschitz with respect to some metric $d_{\mathcal{X}}$ on $\mathcal{X}$, then it follows that $f$ is Lipschitz with constant $\max\{2\|f_0\|_{\mathcal{X}}, L_0\}$ with respect to the product metric

$$d_{\mathcal{Z}}(z, z') = d_{\mathcal{Z}}((x, y), (x', y')) := d_{\mathcal{X}}(x, x') + \mathbf{1}\{y \neq y'\}.$$

Next we consider the case when the function $f$ is smooth but not Lipschitz-continuous. Since we are working with general metric spaces that may lack an obvious differentiable structure, we need to first introduce some concepts from metric geometry [1]. A metric space $(\mathcal{Z}, d_{\mathcal{Z}})$ is a *geodesic space* if for every two points $z, z' \in \mathcal{Z}$ there exists a path $\gamma : [0, 1] \to \mathcal{Z}$, such that $\gamma(0) = z$, $\gamma(1) = z'$, and $d_{\mathcal{Z}}(\gamma(s), \gamma(t)) = (t - s) \cdot d_{\mathcal{Z}}(\gamma(0), \gamma(1))$ for all $0 \leq s \leq t \leq 1$ (such a path is called a *constant-speed geodesic*). A functional $F : \mathcal{Z} \to \mathbb{R}$ is *geodesically convex* if for any pair of points $z, z' \in \mathcal{Z}$ there is a constant-speed geodesic $\gamma$, so that

$$F(\gamma(t)) \leq (1 - t) F(\gamma(0)) + t F(\gamma(1)) = (1 - t) F(z) + t F(z'), \qquad \forall t \in [0, 1].$$

An *upper gradient* of a Borel function $f : \mathcal{Z} \to \mathbb{R}$ is a functional $G_f : \mathcal{Z} \to \mathbb{R}_+$, such that for any pair of points $z, z' \in \mathcal{Z}$ there exists a constant-speed geodesic $\gamma$ obeying

$$|f(z') - f(z)| \leq \int_0^1 G_f(\gamma(t)) \mathrm{d}t \cdot d_{\mathcal{Z}}(z, z'). \tag{5}$$

With these definitions at hand, we have the following:

**Proposition 2.** *Suppose that $f$ has a geodesically convex upper gradient $G_f$. Then*

$$R(Q, f) \leq R_{\varrho,p}(P, f) \leq R(Q, f) + 2\varrho \sup_{Q \in B_{\varrho,p}^W(P)} \|G_f(Z)\|_{L^q(Q)},$$

*where $1/p + 1/q = 1$, and $\| \cdot \|_{L^q(Q)} := (\mathbf{E}_Q| \cdot |^q)^{1/q}$.*

Consider the setting of *regression with quadratic loss*: let $\mathcal{X}$ be a convex subset of $\mathbb{R}^d$, let $\mathcal{Y} = [-B, B]$ for some $0 < B < \infty$, and equip $\mathcal{Z} = \mathcal{X} \times \mathcal{Y}$ with the Euclidean metric

$$d_{\mathcal{Z}}(z, z') = \sqrt{\|x - x'\|_2^2 + |y - y'|^2}, \quad z = (x, y), z' = (x', y'). \tag{6}$$

Suppose that the functions $f \in \mathcal{F}$ are of the form $f(z) = f(x, y) = (y - h(x))^2$ with $h \in C^1(\mathbb{R}^d, \mathbb{R})$, such that $\|h\|_{\mathcal{X}} \leq M < \infty$ and $\|\nabla h(x)\|_2 \leq L\|x\|_2$ for some $0 < L < \infty$. Then Proposition 2 leads to the following:

**Proposition 3.**

$$R(Q, f) \leq R_{\varrho, 2}(P, f) \leq R(Q, f) + 4\varrho(B + M)\Big(1 + L \sup_{Q \in B^W_{\varrho, 2}(P)} \sigma_{Q, X}\Big),$$

*where* $\sigma_{Q, X} := \mathbf{E}_Q \|X\|_2$ *for* $Z = (X, Y) \sim Q$.

## 2.2 An illustrative example: $\varrho$ as an exploratory budget

Before providing formal theoretical guarantees for ERM based on the local minimax risk $R_{\varrho, p}(P_n, f)$ in Section 3, we give a stylized yet insightful example to illustrate the key difference between the ordinary ERM and the local minimax ERM. In a nutshell, the local minimax ERM utilizes the Wasserstein radius $\varrho$ as an *exploratory budget* to reject hypotheses overly sensitive to domain drift.

Consider $Z \sim \text{Unif}[0, 1] =: P$ on data space $\mathcal{Z} = [0, 2]$, along with the hypothesis class $\mathcal{F}$ with only two hypotheses:

$$f_0(z) = 1, \quad f_1(z) = \begin{cases} 0, & z \in [0, 1) \\ \alpha, & z \in [1, 2] \end{cases}$$

for some $\alpha \gg 1$. Notice that, if the training data are drawn from $Z$, the ordinary ERM will always return $f_1$, the hypothesis that is not robust against small domain drifts, while we are looking for a structured procedure that will return $f_0$, a hypothesis that works well for probability distributions 'close' to the data-generating distribution $\text{Unif}[0, 1]$.

The success of minimax learning depends solely on the ability to *transport* some weight from a nearby training sample to 1, the region where nonrobust $f_1$ starts to perform poorly. Specifically, the minimax learning is 'successful' when $R_{\varrho, p}(P_n, f_0) = 1$ is smaller than $R_{\varrho, p}(P_n, f_1) \approx \alpha \varrho^p / (1 - \max Z_i)^p$, which happens with probability $1 - (1 - \varrho \alpha^{1/p})^n$.

We make following key observations:

- While smaller $\varrho$ leads to the smaller nontrivial excess risk $R_{\varrho, p}(P, f_1) - R_{\varrho, p}(P, f_0)$, it also leads to a *slower* decay of error probability. As a result, for a given $\varrho$, we can come up with a hypothesis class maximizing the excess risk at target $\varrho$ with excess risk behaving roughly as $\varrho^{-p^2/(p+1)}$ without affecting the Rademacher average of the class (see supplementary Appendix B for details).

- It is possible to guarantee smooth behavior of the ERM hypothesis without having uniform smoothness assumptions on $\mathcal{F}$; if there exists a single smooth hypothesis $f_0$, it can be used as a baseline comparison to reject nonsmooth hypotheses. We build on this idea in Section 3.3.

## 3 Guarantees for empirical risk minimization

Let $Z_1, \ldots, Z_n$ be an $n$-tuple of i.i.d. training examples drawn from $P$. In this section, we analyze the performance of the *local minimax ERM* procedure

$$\widehat{f} := \underset{f \in \mathcal{F}}{\arg \min} \, R_{\varrho, p}(P_n, f). \tag{7}$$

The following strong duality result due to Gao and Kleywegt [11] will be instrumental:

**Proposition 4.** *For any upper semicontinuous function* $f : \mathcal{Z} \to \mathbb{R}$ *and for any* $Q \in \mathcal{P}_p(\mathcal{Z})$,

$$R_{\varrho, p}(Q, f) = \min_{\lambda \geq 0} \left\{ \lambda \varrho^p + \mathbf{E}_Q[\varphi_{\lambda, f}(Z)] \right\}, \tag{8}$$

*where* $\varphi_{\lambda, f}(z) := \sup_{z' \in \mathcal{Z}} \left\{ f(z') - \lambda \cdot d^p_{\mathcal{Z}}(z, z') \right\}$.

### 3.1 Data-dependent bound on generalization error

We begin by imposing standard regularity assumptions (see, e.g., [7]) which allow us to invoke concentration-of-meausre results for empirical processes.

**Assumption 1.** *The instance space* $\mathcal{Z}$ *is bounded:* $\text{diam}(\mathcal{Z}) := \sup_{z, z' \in \mathcal{Z}} d_{\mathcal{Z}}(z, z') < \infty$.

**Assumption 2.** *The functions in $\mathcal{F}$ are upper semicontinuous and uniformly bounded: $0 \leq f(z) \leq M < \infty$ for all $f \in \mathcal{F}$ and $z \in \mathcal{Z}$.*

As a complexity measure of the hypothesis class $\mathcal{F}$, we use the *entropy integral* [19]

$$\mathfrak{C}(\mathcal{F}) := \int_0^\infty \sqrt{\log \mathcal{N}(\mathcal{F}, \|\cdot\|_\infty, u)} \mathrm{d}u,$$

where $\mathcal{N}(\mathcal{F}, \|\cdot\|_\infty, \cdot)$ denotes the covering number of $\mathcal{F}$ in the uniform metric $\|f - f'\|_\infty = \sup_{z \in \mathcal{Z}} |f(z) - f'(z)|$.

The benefits of using the entropy integral $\mathfrak{C}(\mathcal{F})$ instead of usual complexity measures such as Rademacher or Gaussian complexity [3] are twofold: (1) $\mathfrak{C}(\mathcal{F})$ takes into account the behavior of hypotheses outside the support of the data-generating distribution $P$, and thus can be applied for the assessment of local worst-case risk; (2) Rademacher complexity of $\varphi_{\lambda,f}$ can be upper-bounded naturally via $\mathfrak{C}(\mathcal{F})$ and the covering number of a suitable bounded subset of $[0, \infty)$.

We are now ready to give our data-dependent bound on $R_{\varrho,p}(P, f)$:

**Theorem 1.** *For any $\mathcal{F}, P$ satisfying Assumptions 1–2 and for any $t > 0$,*

$$\mathbf{P}\Bigg( \exists f \in \mathcal{F} : R_{\varrho,p}(P, f) > \min_{\lambda \geq 0} \Bigg\{ (\lambda + 1)\varrho^p + \mathbf{E}_{P_n}[\varphi_{\lambda,f}(Z)] + \frac{M\sqrt{\log(\lambda+1)}}{\sqrt{n}} \Bigg\}$$
$$+ \frac{24\mathfrak{C}(\mathcal{F})}{\sqrt{n}} + \frac{Mt}{\sqrt{n}} \Bigg) \leq 2 \exp(-2t^2)$$

*and*

$$\mathbf{P}\Bigg( \exists f \in \mathcal{F} : R_{\varrho,p}(P_n, f) > \min_{\lambda \geq 0} \Bigg\{ (\lambda + 1)\varrho^p + \mathbf{E}_P[\varphi_{\lambda,f}(Z)] + \frac{M\sqrt{\log(\lambda+1)}}{\sqrt{n}} \Bigg\}$$
$$+ \frac{24\mathfrak{C}(\mathcal{F})}{\sqrt{n}} + \frac{Mt}{\sqrt{n}} \Bigg) \leq 2 \exp(-2t^2).$$

Notice that Theorem 1 is in the style of data-dependent generalization bounds for *margin cost function* class [14], often used for the analysis of voting methods or support vector machines [2].

**Remark 2.** When $\varrho = 0$, we recover the behavior of the usual statistical risk $R(P, f)$. Specifically, it is not hard to show from the definition of $\varphi_{\lambda,f}$ that $\mathbf{E}_{P_n}[\varphi_{\lambda,f}] = \mathbf{E}_{P_n}[f]$ holds for all

$$\lambda \geq \widehat{\lambda}_n := \max_{1 \leq i \leq n} \sup_{z' \in \mathcal{Z}} \frac{f(z') - f(Z_i)}{d_{\mathcal{Z}}^p(z', Z_i)}.$$

In that case, when $\varrho = 0$, the generalization error converges to zero at the rate of $1/\sqrt{n}$ with usual coefficients from the Dudley's entropy integral [19] and McDiarmid's inequality, plus an added term of order $\frac{M\sqrt{\log \lambda^*}}{\sqrt{n}}$ for some $\lambda^* \geq \widehat{\lambda}_n$.

### 3.2 Excess risk bounds with uniform smoothness

As evident from Remark 2, if we have *a priori* knowledge that the hypothesis selected by the minimax ERM procedure (7) is *smooth* with respect to the underlying metric, then we can restrict the feasible values of $\lambda$ to provide data-independent guarantees on generalization error, which vanishes to 0 as $n \to \infty$. Let us start by imposing the following 'uniform smoothness' on $\mathcal{F}$:

**Assumption 3.** *The functions in $\mathcal{F}$ are $L$-Lipschitz:* $\sup_{z \neq z'} \frac{f(z') - f(z)}{d_{\mathcal{Z}}(z', z)} \leq L$ *for all $f \in \mathcal{F}$.*

One motivation for Assumption 3 is the following bound on the excess risk: whenever the solution of the original ERM $\tilde{f} = \arg\min_{f \in \mathcal{F}} \sum_{i=1}^n f(z_i)$ is $L$-Lipschitz, Kantorovich duality gives us

$$R_{\varrho,p}(P, \tilde{f}) - R^*_{\varrho,p}(P, \mathcal{F}) \leq R(P, \tilde{f}) - R^*(P, \mathcal{F}) + L\varrho, \qquad (9)$$

where the right-hand side is the sum of excess risk of ordinary ERM, and the worst-case deviation of risk due to the ambiguity. The bound (9) is particularly useful when both $\varrho$ is and $n$ are small, but it does not vanish as $n \to \infty$.

The following lemma enables the control of *infimum-achieving* dual parameter $\lambda$ with respect to the true and empirical distribution:

**Lemma 1.** *Fix some $Q \in \mathcal{P}_p(\mathcal{Z})$, and define $\tilde{f} \in \mathcal{F}$ and $\tilde{\lambda} \geq 0$ via*

$$\tilde{f} := \underset{f \in \mathcal{F}}{\arg\min}\, R_{\varrho,p}(Q, f) \quad and \quad \tilde{\lambda} := \underset{\lambda \geq 0}{\arg\min}\left\{\lambda\varrho^p + \mathbf{E}_Q[\varphi_{\lambda,\tilde{f}}(Z)]\right\}.$$

*Then under Assumptions 1–3, we have $\tilde{\lambda} \leq L\varrho^{-(p-1)}$.*

Then, we can use the Dudley entropy integral arguments [19] on the joint search space of $\lambda$ and $f$ to get the following theorem:

**Theorem 2.** *Under Assumptions 1–3, the following holds with probability at least $1 - \delta$:*

$$R_{\varrho,p}(P, \widehat{f}) - R_{\varrho,p}^*(P, \mathcal{F}) \leq \frac{48\mathfrak{C}(\mathcal{F})}{\sqrt{n}} + \frac{48L \cdot \mathsf{diam}(\mathcal{Z})^p}{\sqrt{n} \cdot \varrho^{p-1}} + 3M\sqrt{\frac{\log(2/\delta)}{2n}}. \tag{10}$$

**Remark 3.** The adversarial training procedure appearing in a concurrent work of Sinha et al. [18] can be interpreted as a relaxed version of local minimax ERM, where we consider $\lambda$ to be fixed (to enhance implementability), rather than explicitly searching for an optimal $\lambda$. In such case, Lemma 1 may provide a guideline for the selection of parameter $\lambda$; for example, one might run the fixed-$\lambda$ algorithm over a sufficiently fine grid of $\lambda$ on the interval $[0, L\varrho^{-(p-1)}]$ to approximate the local minimax ERM.

Note that when $p = 1$, we get a $\varrho$-free bound of order $1/\sqrt{n}$, recovering the correct rate of ordinary ERM as $\varrho = 0$. On the other hand, Theorem 2 cannot be used to recover the rate of ordinary ERM for $p > 1$. This phenomenon is due to the fact that we are using the Lipschitz assumption on $\mathcal{F}$, which is a *data-independent* constraint on the scale of the trade-off between $f(z') - f(z)$ and $d_{\mathcal{Z}}(z', z)$. For $p > 1$, one may also think of a similar *data-independent* (or, worst-case) constraint

$$\sup_{z,z'} \frac{f(z') - f(z)}{d_{\mathcal{Z}}^p(z', z)} < +\infty.$$

However, this holds only if $f$ is constant, even in the simplest case $\mathcal{Z} \subseteq \mathbb{R}$.

### 3.3 Excess risk bound with minimal assumptions

The illustrative example presented in Section 2.2 implies that the minimax learning might be possible even when the functions in $\mathcal{F}$ are not uniformly Lipschitz, but there exists at least one smooth hypothesis (at least, except for the regime $\varrho \to 0$). Based on that observation, we now consider a weaker alternative to Assumption 3:

**Assumption 4.** *There exists a hypothesis $f_0 \in \mathcal{F}$, such that, for all $z \in \mathcal{Z}$, $f_0(z) \leq C_0 d_{\mathcal{Z}}^p(z, z_0)$ for some $C_0 \geq 0$ and $z_0 \in \mathcal{Z}$.*

Assumption 4 guarantees the existence of a hypothesis with smooth behavior with respect to the underlying metric $d_{\mathcal{Z}}$; on the other hand, smoothness is not required for every $f \in \mathcal{F}$, and thus Assumption 4 is particularly useful when paired with a rich class $\mathcal{F}$.

It is not difficult to see that Assumption 4 holds for most common hypothesis classes. As an example, consider again the setting of *regression with quadratic loss* as in Proposition 3; the functions $f \in \mathcal{F}$ are of the form $f(z) = f(x, y) = (y - h(x))^2$, where $h$ runs over some given class of candidate predictors that contains constants. Then, we can take $h_0(x) \equiv 0$, in which case $f_0(z) = (h_0(x) - y)^2 = |y|^2 \leq d_{\mathcal{Z}}^2(z, z_0)$ for all $z_0$ of the form $(x, 0) \in \mathcal{X} \times \mathcal{Y}$.

Under Assumption 4, we can prove the following counterpart of Lemma 1:

**Lemma 2.** *Fix some $Q \in \mathcal{P}_p(\mathcal{Z})$. Define $\tilde{f} \in \mathcal{F}$ and $\tilde{\lambda} \geq 0$ via*

$$\tilde{f} := \underset{f \in \mathcal{F}}{\arg\min}\, R_{\varrho,p}(Q, f) \quad and \quad \tilde{\lambda} := \underset{\lambda \geq 0}{\arg\min}\left\{\lambda\varrho^p + \mathbf{E}_Q[\varphi_{\lambda,\tilde{f}}(Z)]\right\}.$$

*Then, under Assumptions 1,2,4, $\tilde{\lambda} \leq C_0 2^{p-1} \left(1 + (\mathsf{diam}(\mathcal{Z})/\varrho)^p\right)$.*

An intuition behind Lemma 2 is to interpret the Wasserstein perturbation $\varrho$ as a regularization parameter to thin out hypotheses with non-smooth behavior around $Q$ by comparing it to $f_0$. As $\varrho$ grows, a smaller dual parameter $\lambda$ is sufficient to control the adversarial behavior.

We can now give a performance guarantee for the ERM procedure (7):

**Theorem 3.** *Under Assumptions 1,2,4, the following holds with probability at least $1 - \delta$:*

$$R_{\varrho,p}(P,\widehat{f}) - R^*_{\varrho,p}(P,\mathcal{F}) \leq \frac{48\mathfrak{C}(\mathcal{F})}{\sqrt{n}} + \frac{24C_0(2\,\mathsf{diam}(\mathcal{Z}))^p}{\sqrt{n}} \left(1 + \left(\frac{\mathsf{diam}(\mathcal{Z})}{\varrho}\right)^p\right) + 3M\sqrt{\frac{\log(2/\delta)}{2n}}.$$
(11)

**Remark 4.** The second term decreases as $\varrho$ grows, which is consistent with the phenomenon illustrated in Section 2.2. Also note that the excess risk bound of [9] shows the same behavior as Theorem 3, where in that case $\varrho$ is the slack in the moment constraints defining the ambiguity set. While larger ambiguity can be helpful for learnability in this sense, note that the risk inequalities of Sec 2.1 imply that $R_{\varrho,p}(P,f) - R(P,f)$ can be bigger with larger $\varrho$. Using these two elements, one can provide domain-specific excess risk bounds which explicitly describe the interplay of both elements with ambiguity (see Sec 4).

### 3.4 Example bounds

In this subsection, we illustrate the use of Theorem 2 when (upper bounds on) the covering numbers for the hypothesis class $\mathcal{F}$ are available. Throughout this section, we continue to work in the setting of regression with quadratic loss as in Proposition 3; we let $\mathcal{X} = \{x \in \mathbb{R}^d : \|x\|_2 \leq r_0\}$ be a ball of radius $r_0$ in $\mathbb{R}^d$ centered at the origin, let $\mathcal{Y} = [-B, B]$ for some $B > 0$, and equip $\mathcal{Z}$ with the Euclidean metric (6). Also, we take $p = 1$.

We first consider a simple neural network class $\mathcal{F}$ consisting of functions of the form $f(z) = f(x,y) = (y - s(f_0^T x))^2$, where $s : \mathbb{R} \to \mathbb{R}$ is a bounded smooth nonlinearity with $s(0) = 0$ and with bounded first derivative, and where $f_0$ takes values in the unit ball in $\mathbb{R}^d$.

**Corollary 1.** *For any $P \in \mathcal{P}(\mathcal{Z})$, with probability at least $1 - \delta$,*

$$R_{\varrho,1}(P,\widehat{f}) - R^*_{\varrho,1}(P,\mathcal{F}) \leq \frac{C_1}{\sqrt{n}} + \frac{3(\|s\|_\infty + B)^2\sqrt{\log(2/\delta)}}{\sqrt{2n}}$$

*where $C_1$ is a constant dependent only on $d, r_0, s, B$:*

$$C_1 = (B + \|s\|_\infty) \cdot \left(144 r_0 \sqrt{d}\|s'\|_\infty + 192(1 + \|s'\|_\infty)\sqrt{2(r_0^2 + B^2)}\right).$$

We also consider the case of a massive nonparametric class. Let $(\mathcal{H}_K, \|\cdot\|_K)$ be the Gaussian reproducing kernel Hilbert space (RKHS) with the kernel $K(x_1, x_2) = \exp\{-\|x_1 - x_2\|_2^2/\sigma^2\}$ for some $\sigma > 0$, and let $\mathcal{B}_r := \{h \in \mathcal{H}_K : \|h\|_K \leq r\}$ be the radius-$r$ ball in $\mathcal{H}_K$. Let $\mathcal{F}$ be the class of all functions of the form $f(z) = f(x,y) = (y - f_0(x))^2$, where the predictors $f_0 : \mathcal{X} \to \mathbb{R}$ belong to $I_K(\mathcal{B}_r)$, an embedding of $\mathcal{B}_r$ into the space $C(\mathcal{X})$ of continuous real-valued functions on $\mathcal{X}$ equipped with the sup norm $\|f\|_{\mathcal{X}} := \sup_{x \in \mathcal{X}} |f(x)|$.

Using the covering number estimates due to Cucker and Zhou [7], we can prove the following generalization bounds for Gaussian RKHS.

**Corollary 2.** *With probability at least $1 - \delta$, for any $P \in \mathcal{P}(\mathcal{Z})$,*

$$R_{\varrho,1}(P,\widehat{f}) - R^*_{\varrho,1}(P,\mathcal{F}) \leq$$
$$\frac{C_1}{\sqrt{n}}(r^2 + Br) + \frac{192\sqrt{2}(r + B) \cdot (1 + r\sqrt{2}/\sigma)\sqrt{r_0^2 + B^2}}{\sqrt{n}} + \frac{6(r^2 + B^2)\sqrt{\log(2/\delta)}}{\sqrt{2n}}$$

*where $C_1$ is a constant dependent only on $d, r_0, \sigma$:*

$$C_1 = 48\sqrt{d}\left(2\Gamma\left(\frac{d+3}{2}, \log 2\right) + (\log 2)^{\frac{d+1}{2}}\right)\left(32 + \frac{2560 d r_0^2}{\sigma^2}\right)^{\frac{d+1}{2}}$$

*(here, $\Gamma(s, v) := \int_v^\infty u^{s-1} e^{-u} \mathrm{d}u$ is the incomplete gamma function).*

# 4 Application: Domain adaptation with optimal transport

Ambiguity sets based on Wasserstein distances have two attractive features. First, the metric geometry of the instance space provides a natural mechanism for handling uncertainty due to transformations on the problem instances. For example, concurrent work by Sinha et al. [18] interprets the underlying metric as a perturbation cost of an adversary in the context of *adversarial examples* [12]. Second, Wasserstein distances can be approximated efficiently from the samples; Fournier and Guillin [10] provide nonasymptotic convergence results in terms of both moments and probability for general $p$. This allows us to approximate the Wasserstein distance between two distributions $W_p(P, Q)$ by the Wasserstein distance between their empirical distributions $W_p(P_n, Q_n)$, which makes it possible to specify a suitable level of ambiguity $\varrho$.

One interesting area of application, where we benefit from both of these aspects is the problem of *domain adaptation*, arising when we want to transfer the data/knowledge from a source domain $P \in \mathcal{P}(\mathcal{Z})$ to a different but related target domain $Q \in \mathcal{P}(\mathcal{Z})$ [4]. While the domain adaptation problem is often stated in a broader context, we confine our discussion to adaptation in supervised learning, assuming $\mathcal{Z} = \mathcal{X} \times \mathcal{Y}$ where $\mathcal{X}$ is the feature space and $\mathcal{Y}$ is the label space. From now on, we disintegrate the source distribution as $P = \mu \otimes P_{Y|X}$ and target distribution as $Q = \nu \otimes Q_{Y|X}$.

Existing theoretical results on domain adaptation are phrased in terms of the 'discrepancy metric' [15]: given a loss function $l : \mathcal{Y} \times \mathcal{Y} \rightarrow \mathbb{R}$ and a family of predictors $\mathcal{H}$ of form $h : \mathcal{X} \rightarrow \mathcal{Y}$, the discrepancy metric is defined as

$$\mathsf{disc}_{\mathcal{H}}(\mu, \nu) := \max_{h, h' \in \mathcal{H}} |\mathbf{E}_{\mu} [l(h(X), h'(X))] - \mathbf{E}_{\nu} [l(h(X), h'(X))]| .$$

Typical theoretical guarantees involving the discrepancy metric take the form of *generalization bounds*: for any $h \in \mathcal{H}$,

$$R(Q, h) - R^*(Q, \mathcal{H}) \leq R(P, h) + \mathsf{disc}_{\mathcal{H}}(\mu, \nu) + \mathbf{E}_{\nu} \left[ l(h_P^*(X), h_Q^*(X)) \right] \tag{12}$$

where $h_P^*$ and $h_Q^*$ are minimizers of $R(P, h) = \mathbf{E}_P[l(h(X), Y)]$ and $R(Q, h) = \mathbf{E}_Q[l(h(X), Y)]$. While these generalization bounds provide a uniform guarantee for all predictors in a class, they can be considered 'pessimistic' in the sense that we compare the excess risk to $R(P, h)$, which is the performance of some selected predictor at the source domain.

Our work, on the other hand, aims to provide an excess risk bound for a specific target hypothesis $\widehat{f}$ given by the solution of a minimax ERM. Suppose that it is possible to estimate the Wasserstein distance $W_p(P, Q)$ between the two domain distributions. Then, as we show below, we can provide a generalization bound for the target domain by combining estimation guarantees for $W_p(P, Q)$ with risk inequalities of Section 2. All proofs are given in supplementary Appendix E.

We work in the setting considered by Courty et al. [6]: Let $\mathcal{X}, \mathcal{Y}$ be metric spaces with metric $d_{\mathcal{X}}$ and $d_{\mathcal{Y}}$. We then endow $\mathcal{Z}$ with the $\ell_p$ product metric

$$d_{\mathcal{Z}}(z, z') = d_{\mathcal{Z}}((x, y), (x', y')) := \left( d_{\mathcal{X}}^p(x, x') + d_{\mathcal{Y}}^p(y, y') \right)^{1/p}.$$

We assume that domain drift is due to an unknown (possibly nonlinear) transformation $T : \mathcal{X} \rightarrow \mathcal{X}$ of the feature space that preserves the conditional distribution of the labels given the features, e.g. acquisition condition, sensor drift, thermal noise, etc. That is, $\nu = T_{\#}\mu$, the pushforward of $\mu$ by $T$, and for any $x \in \mathcal{X}$ and any measurable set $B \subseteq \mathcal{Y}$

$$P_{Y|X}(B|x) = Q_{Y|X}(B|T(x)). \tag{13}$$

This assumption leads to the following lemma, which enables us to estimate $W_p(P, Q)$ only from unlabeled source domain data and unlabeled target domain data:

**Lemma 3.** *Suppose there exists a deterministic and invertible optimal transport map $T : \mathcal{X} \rightarrow \mathcal{X}$ such that $\nu = T_{\#}\mu$, i.e., $W_p^p(\mu, \nu) = \mathbf{E}_{\mu}[d_{\mathcal{X}}^p(X, T(X))]$. Then*

$$W_p(P, Q) = W_p(\mu, \nu). \tag{14}$$

**Remark 5.** If $\mathcal{X}$ is a convex subset of $\mathbb{R}^d$ endowed with the $\ell_p$ metric $d_{\mathcal{X}}(x, x') = \|x - x'\|_p$ for $p \geq 2$, then, under the assumption that $\mu$ and $\nu$ have positive densities with respect to the Lebesgue measure, the (unique) optimal transport map from $\mu$ to $\nu$ is deterministic and a.e. invertible – in fact, its inverse is equal to the optimal transport map from $\nu$ to $\mu$ [21]. $\qquad \square$

Now suppose that we have $n$ labeled examples $(X_1, Y_1), \ldots, (X_n, Y_n)$ from $P$ and $m$ unlabeled examples $X'_1, \ldots, X'_m$ from $\nu$. Define the empirical distributions

$$\mu_n = \frac{1}{n} \sum_{i=1}^{n} \delta_{X_i}, \qquad \nu_m = \frac{1}{m} \sum_{j=1}^{m} \delta_{X'_j}.$$

Notice that, by the triangle inequality, we have

$$W_p(\mu, \nu) \leq W_p(\mu, \mu_n) + W_p(\mu_n, \nu_m) + W_p(\nu, \nu_m). \tag{15}$$

Here, $W_p(\mu_n, \nu_m)$ can be computed from unlabeled data by solving a finite-dimensional linear program [21], and the following convergence result of Fournier and Guillin [10] implies that, with high probability, both $W_p(\mu, \mu_n)$ and $W_p(\nu, \nu_m)$ rapidly converge to zero as $n, m \to \infty$:

**Proposition 5.** *Let $\mu$ be a probability distribution on a bounded set $\mathcal{X} \subset \mathbb{R}^d$, where $d > 2p$. Let $\mu_n$ denote the empirical distribution of $X_1, \ldots, X_n \overset{\text{i.i.d.}}{\sim} \mu$. Then, for any $r \in (0, \infty)$,*

$$\mathbf{P}(W_p(\mu_n, \mu) \geq r) \leq C_a \exp(-C_b n r^{d/p}) \tag{16}$$

*where $C_a, C_b$ are constants depending on $p, d, \mathsf{diam}(\mathcal{X})$ only.*

**Remark 6.** Note that $d > 2p$ is not a necessary constraint, and the bound still holds in the case $d \leq 2p$ with different speed of convergence. In particular, Proposition 5 is a constrained version of [10, Thm. 2] under finite $\mathcal{E}_{\alpha,\gamma}(\mu)$ for $\alpha = d > p$. $\qquad\square$

Based on these considerations, we propose the following domain adaptation scheme:

1. Compute the $p$-Wasserstein distance $W_p(\mu_n, \nu_m)$ between the empirical distributions of the features in the labeled training set from the source domain $P$ and the unlabeled training set from the target domain $Q$.

2. Set the desired confidence parameter $\delta \in (0, 1)$ and the radius

$$\widehat{\varrho}(\delta) := W_p(\mu_n, \nu_m) + \left( \frac{\log(4C_a/\delta)}{C_b n} \right)^{p/d} + \left( \frac{\log(4C_a/\delta)}{C_b m} \right)^{p/d}. \tag{17}$$

3. Compute the empirical risk minimizer

$$\widehat{f} = \underset{f \in \mathcal{F}}{\arg\min}\, R_{\widehat{\varrho}(\delta), p}(P_n, f), \tag{18}$$

   where $P_n$ is the empirical distribution of the $n$ labeled samples from $P$.

We can give the following target domain generalization bound for the hypothesis generated by (18):

**Theorem 4.** *Suppose that the feature space $\mathcal{X}$ is a bounded subset of $\mathbb{R}^d$ with $d > 2p$, take $d_{\mathcal{X}}(x, x') = \|x - x'\|_p$, and let $\mathcal{F}$ be a family of hypotheses with Lipschitz constant at most $L$. Then, the empirical risk minimizer $\widehat{f}$ from (18) satisfies*

$$R(Q, \widehat{f}) - R^*(Q, \mathcal{F}) \leq 2L\widehat{\varrho}(\delta) + \frac{48\mathfrak{C}(\mathcal{F})}{\sqrt{n}} + \frac{48L \cdot \mathsf{diam}^p(\mathcal{Z})}{\sqrt{n}\widehat{\varrho}^{p-1}} + \frac{3M\sqrt{\log(4/\delta)}}{\sqrt{2n}}.$$

*with probability at least $1 - \delta$.*

**Remark 7.** Comparing the bound of Theorem 4 with the discrepancy-based bound (12), we note that the former does not contain any terms related to $R(P, \widehat{f})$ or the closeness of the optimal predictors for $P$ and $Q$. The only contributions to the excess risk are the empirical Wasserstein distance $W_p(\mu_n, \nu_m)$ (which captures the discrepancy between the source and the target domains in a data-driven manner) and an empirical process fluctuation term. In this sense, the bound of Theorem 4 is closer in spirit to the usual excess risk bounds one obtains in the absence of domain drift. $\qquad\square$

## Acknowledgement

We would like to thank Pierre Moulin, Yung Yi, and anonymous reviewers for helpful discussions.

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
