[Supplementary Material]

# A Proofs for Section 2

## A.1 Proof of Proposition 1

For $p = 1$, the result follows immediately from the Kantorovich dual representation of $W_1(\cdot, \cdot)$ [21]:

$$W_1(Q, Q') = \sup \left\{ |\mathbf{E}_Q F - \mathbf{E}_{Q'} F| : \sup_{\substack{z, z' \in \mathcal{Z} \\ z \neq z'}} \frac{|F(z) - F(z')|}{d_{\mathcal{Z}}(z, z')} \leq 1 \right\}$$

and from the fact that, for $Q, Q' \in B^W_{\varrho,1}(P)$, $W_1(Q, Q') \leq 2\varrho$ by the triangle inequality. For $p > 1$, the result follows from the fact that $W_1(Q, Q') \leq W_p(Q, Q')$ for all $Q, Q' \in \mathcal{P}_p(\mathcal{Z})$.

## A.2 Proof of Proposition 2

Fix some $Q, Q' \in B^W_{\varrho,p}(P)$ and let $M \in \mathcal{P}(\mathcal{Z} \times \mathcal{Z})$ achieve the infimum in (4) for $W_p(Q, Q')$. Then for $(Z, Z') \sim M$ we have

$$f(Z') - f(Z) \leq \int_0^1 G_f(\gamma(t)) \mathrm{d}t \cdot d_{\mathcal{Z}}(Z, Z')$$
$$\leq \frac{1}{2} \left( G_f(Z) + G_f(Z') \right) d_{\mathcal{Z}}(Z, Z'),$$

where the first inequality is from (5) and the second one is by the assumed geodesic convexity of $G_f$. Taking expectations of both sides with respect to $M$ and using Hölder's inequality, we obtain

$$R(Q', f) - R(Q, f) \leq \frac{1}{2} \left( \mathbf{E}_M |G_f(Z) + G_f(Z')|^q \right)^{1/q} \left( \mathbf{E}_M d^p_{\mathcal{Z}}(Z, Z') \right)^{1/p}$$
$$= \frac{1}{2} \| G_f(Z) + G_f(Z') \|_{L^q(M)} W_p(Q, Q'),$$

where we have used the $p$-Wasserstein optimality of $M$ for $Q$ and $Q'$. By the triangle inequality, and since $Z \sim Q$ and $Z' \sim Q'$,

$$\| G_f(Z) + G_f(Z') \|_{L^q(M)} \leq \| G_f(Z) \|_{L^q(Q)} + \| G_f(Z) \|_{L^q(Q')}$$
$$\leq 2 \sup_{Q \in B^W_{\varrho,p}(P)} \| G_f(Z) \|_{L^q(Q)}.$$

Interchanging the roles of $Q$ and $Q'$ and proceeding with the same argument, we obtain the estimate

$$\sup_{Q, Q' \in B^W_{\varrho,p}(P)} |R(Q, f) - R(Q', f)| \leq 2\varrho \sup_{Q \in B^W_{\varrho,p}(P)} \| G_f(Z) \|_{L^q(Q)},$$

from which it follows that

$$R(Q, f) \leq R_{\varrho,p}(P, f)$$
$$= \sup_{Q' \in B^W_{\varrho,p}(P)} [R(Q', f) - R(Q, f) + R(Q, f)]$$
$$\leq R(Q, f) + 2\varrho \sup_{Q \in B^W_{\varrho,p}(P)} \| G_f(Z) \|_{L^q(Q)}.$$

## A.3 Proof of Proposition 3

As a subset of $\mathbb{R}^{d+1}$, $\mathcal{Z}$ is a geodesic space: for any pair $z, z' \in \mathcal{Z}$ there is a unique constant-speed geodesic $\gamma(t) = (1-t)z + tz'$. We claim that $G_f(z) = G_f(x, y) = 2(B + M)(1 + L\|\nabla h(x)\|_2)$ is a geodesically convex upper gradient for $f(z) = f(x, y) = (y - h(x))^2$. In this flat Euclidean setting, geodesic convexity coincides with the usual definition of convexity, and the map $z \mapsto G_f(z)$ is evidently convex:

$$G_f((1-t)z + tz') \leq (1-t)G_f(z) + tG_f(z').$$

Next, by the mean-value theorem,

$$f(z') - f(z) = \int_0^1 \langle z' - z, \nabla f((1-t)z + t'z)\rangle \mathrm{d}t$$

$$\leq \int_0^1 \|\nabla f((1-t)z + tz')\|_2 \, \mathrm{d}t \cdot \|z - z'\|_2$$

$$= \int_0^1 \|\nabla f((1-t)z + tz')\|_2 \, \mathrm{d}t \cdot d_{\mathcal{Z}}(z, z'),$$

and a simple calculation shows that

$$\|\nabla f(z)\|_2^2 = \|\nabla f(x, y)\|_2^2$$
$$= 4f(z)\left(1 + \|\nabla h(z)\|_2^2\right)$$
$$\leq 4(B + M)^2(1 + L^2\|x\|_2^2).$$

Therefore, $\|\nabla f(z)\|_2 \leq G_f(z)$ for $z = (x, y)$, as claimed. Thus, by Proposition (2),

$$R(Q, f) \leq R_{\varrho,2}(P, f)$$
$$\leq R(Q, f) + 2 \sup_{Q \in B_{\varrho,2}^W(P)} \|G_f(Z)\|_{L^2(Q)}\varrho$$
$$= R(Q, f) + 4(B + M)\left(1 + L \sup_{Q \in B_{\varrho,2}^W(P)} \mathbf{E}_Q\|X\|_2\right)\varrho$$
$$= R(Q, f) + 4(B + M)\left(1 + L \sup_{Q \in B_{\varrho,2}^W(P)} \sigma_{Q,X}\right)\varrho.$$

# B    The illustrative example of Section 2.2

Consider $Z \sim \mathrm{Unif}[0, 1] =: P$ on data space $\mathcal{Z} = [0, 2]$, along with the hypothesis class $\mathcal{F}$ with only two hypotheses

$$f_0(z) = 1, \quad f_1(z) = \begin{cases} 0, & z \in [0, 1) \\ \alpha, & z \in [1, 2] \end{cases},$$

for some constant $\alpha \gg 1$. Also, let $d_{\mathcal{Z}}(z, z') = |z - z'|$.

Now we calculate the local minimax risk of the hypothesis class for both empirical and population measure. The local worst-case risk of $f_0$ for both measures is 1, by definition. For $f_1$, it is easy to see that the worst-case distribution for both $P$ and $P_n$ can be specified explicitly: For $P$, it is optimal to transport the mass to the point $z = 1$ from the interval $[\beta, 1)$, with $\beta \in [0, 1)$ specified according to the ambiguity radius $\varrho > 0$. For $P$, the optimal $\beta$ can be calculated as a solution to

$$\left(\int_\beta^1 (1-z)^p \mathrm{d}z\right)^{1/p} = \varrho,$$

which gives $\beta = 1 - (p+1)^{\frac{1}{p+1}}\varrho^{\frac{p}{p+1}}$, leading to the local worst-case risk $R_{\varrho,p}(P, f_1) = \alpha \cdot (p+1)^{\frac{1}{p+1}}\varrho^{\frac{p}{p+1}}$. For $P_n$, it is optimal to transport mass from the largest value among the training data $\{Z_i\}_{i=1}^n$ to $z = 1$, where the amount of mass $\gamma$ to be transported is given as a solution to

$$\left(\gamma \cdot (1 - \max_{i \in [n]} Z_i)^p\right)^{1/p} = \varrho,$$

which gives $\gamma = \frac{\varrho^p}{(1 - \max Z_i)^p}$ leading to the local worst-case risk of $R_{\varrho,p}(P_n, f_1) = \frac{\alpha\varrho^p}{(1 - \max Z_i)^p}$[2].

The outcome of the local minimax ERM procedure is the hypothesis minimizing the local worst-case risk. In other words, the local minimax hypothesis is $f_0$ whenever $1 \leq \frac{\alpha \varrho^p}{(1-\max Z_i)^p}$ holds, which is true whenever $1 - \alpha^{1/p}\varrho < \max Z_i$. Thus, the local minimax ERM procedure gives

$$\widehat{f} = \begin{cases} f_1, & \text{with probability } \left(1 - \varrho\alpha^{\frac{1}{p}}\right)^n, \\ f_0 & \text{otherwise} \end{cases}$$

for any $\varrho \leq \alpha^{-1/p}$. On the other hand, the minimizer of the local minimax risk with respect to $P$ is given by

$$f^* = \begin{cases} f_1, & \varrho \leq (p+1)^{-\frac{1}{p}}\alpha^{-\frac{p+1}{p}} \\ f_0, & \varrho \geq (p+1)^{-\frac{1}{p}}\alpha^{-\frac{p+1}{p}} \end{cases}$$

Now if we calculate the excess risk of the local minimax ERM hypothesis, we get

$$R_{\varrho,p}(\widehat{f},P) - R_{\varrho,p}(f^*,P) = \begin{cases} \alpha(p+1)^{\frac{1}{p+1}}\varrho^{\frac{p}{p+1}} - 1 & \text{w.p. } \left(1 - \varrho\alpha^{\frac{1}{p}}\right)^n, \\ 0, & \text{otherwise} \end{cases}$$

for any $\varrho \in [(p+1)^{-1/p}\alpha^{-1-1/p}, \alpha^{-1/p}]$, which is a nonempty interval as $\alpha > 1$. Now, if we look at the quantity

$$\varepsilon_\delta^*(\varrho) := \inf\left\{\varrho \geq 0 \; : \; P\left[R_{\varrho,p}(\widehat{f},P) - R_{\varrho,p}(f^*,P) > \varepsilon\right] < \delta\right\}$$

for some fixed $\delta > 0$, then we get

$$\varepsilon_\delta^*(\varrho) = \begin{cases} \alpha(p+1)^{\frac{1}{p+1}}\varrho^{\frac{p}{p+1}} - 1, & (p+1)^{-\frac{1}{p}}\alpha^{-\frac{p+1}{p}} \leq \varrho \leq (1-\delta^{\frac{1}{n}})\alpha^{-\frac{1}{p}} \\ 0, & \text{otherwise.} \end{cases}$$

Now observe that for some fixed $\varrho$ and $\delta$, we can select $\alpha = (1 - \delta^{1/n})^p\varrho^{-p}$ to incur a nontrivial excess risk of order $\varrho^{-\frac{p^2}{p+1}}$. Moreover, this 'selection of worst $\alpha$' can be done without changing the value of the Rademacher average of $\mathcal{F}$, as $f_1$ does not change on the support of $P$.

## C  Proofs for Section 3

### C.1  Proof of Theorem 1

The proof uses a modification of the techniques of Koltchinskii and Panchenko [14]. From the definition of the local minimax risk, we have, for any $f \in \mathcal{F}$

$$R_{\varrho,p}(P,f) = \min_{\lambda \geq 0}\{\lambda\varrho^p + \mathbf{E}_P[\varphi_{\lambda,f}]\}$$

$$\leq \min_{\lambda \geq 0}\left\{\lambda\varrho^p + \mathbf{E}_{P_n}[\varphi_{\lambda,f}] + \sup_{f \in \mathcal{F}}\left(\mathbf{E}_P[\varphi_{\lambda,f}] - \mathbf{E}_{P_n}[\varphi_{\lambda,f}]\right)\right\},$$

where

$$X_\lambda := \sup_{f \in \mathcal{F}}\left(\mathbf{E}_P[\varphi_{\lambda,f}] - \mathbf{E}_{P_n}[\varphi_{\lambda,f}]\right) = \frac{1}{n}\sup_{f \in \mathcal{F}}\left[\sum_{i=1}^{n}(\mathbf{E}\varphi_{\lambda,f}(Z) - \varphi_{\lambda,f}(Z_i))\right]$$

is a data-dependent random variable for each $\lambda \geq 0$. Since $\varphi_{\lambda,f}(Z_i) \in [0,M]$, we know from McDiarmid's inequality that, for any fixed $\lambda \geq 0$,

$$\mathbf{P}\left(X_\lambda > \mathbf{E}X_\lambda + \frac{Mt}{\sqrt{n}}\right) \leq \exp(-2t^2).$$

Furthermore, using a standard symmetrization argument, we have

$$\mathbf{E}X_\lambda \leq 2 \cdot \mathbf{E}\sup_{f \in \mathcal{F}}\frac{1}{n}\sum_{i=1}^{n}\varepsilon_i\varphi_{\lambda,f}(Z_i)$$

where $\varepsilon_1, \ldots, \varepsilon_n$ are i.i.d. Rademacher random variables independent of $Z_1, \ldots, Z_n$. The $\mathcal{F}$-indexed process $Y = (Y_f)_{f \in \mathcal{F}}$ defined via

$$Y_f = \frac{1}{\sqrt{n}} \varepsilon_i \varphi_{\lambda,f}(Z_i)$$

is clearly zero-mean and subgaussian with respect to the metric $\|f - f'\|_\infty$, as

$$
\begin{aligned}
\mathbf{E}[\exp(t(Y_f - Y_{f'}))] &= \mathbf{E}\left[ \exp\left( \frac{t}{\sqrt{n}} \sum_{i=1}^n \varepsilon_i(\varphi_{\lambda,f}(Z_i) - \varphi_{\lambda,f'}(Z_i)) \right) \right] \\
&= \left( \mathbf{E}\left[ \exp\left( \frac{t}{\sqrt{n}} \varepsilon_1 \cdot \sup_{z'} \inf_{z''}\{f(z') - \lambda d_{\gtrsim}^p(Z_1, z') - f'(z'') + \lambda d_{\gtrsim}^p(Z_1, z'')\} \right) \right] \right)^n \\
&\leq \left( \mathbf{E}\left[ \exp\left( \frac{t}{\sqrt{n}} \varepsilon_1 \cdot \sup_{z'}\{f(z') - f'(z')\} \right) \right] \right)^n \\
&\leq \exp\left( \frac{t^2 \|f - f'\|_\infty^2}{2} \right),
\end{aligned}
$$

where the second line is by independence and last line is by Hoeffding's lemma. Invoking Dudley's entropy integral [19], we get

$$\mathbf{E}X_\lambda \leq \frac{24}{\sqrt{n}} \mathfrak{C}(\mathcal{F})$$

for any $\lambda \geq 0$. Summing up, we have, for any fixed $\lambda \geq 0$,

$$\mathbf{P}\left( \exists f \in \mathcal{F} : R_{\varrho,p}(P, f) > \lambda \varrho^p + \mathbf{E}_{P_n}[\varphi_{\lambda,f}] + \frac{24}{\sqrt{n}} \mathfrak{C}(\mathcal{F}) + \frac{Mt}{\sqrt{n}} \right) \leq \exp(-2t^2).$$

Now, pick the sequences $\lambda_k = k$ and $t_k = t + \sqrt{\log k}$ for $k = 1, 2, 3, \ldots$. Then, by the union bound,

$$
\begin{aligned}
&\mathbf{P}\left( \exists f \in \mathcal{F} : R_{\varrho,p}(P, f) > \min_{k=1,2,\ldots}\left\{ \lambda_k \varrho^p + \mathbf{E}_{P_n}[\varphi_{\lambda_k,f}] + \frac{24}{\sqrt{n}} \mathfrak{C}(\mathcal{F}) + \frac{Mt_k}{\sqrt{n}} \right\} \right) \\
&\leq \sum_{k=1}^\infty \exp(-2t_k^2) \\
&\leq \exp(-2t^2) \sum_{k=1}^\infty \exp(-2\log k) \\
&\leq 2\exp(-2t^2).
\end{aligned}
$$

On the other hand,

$$
\begin{aligned}
&\min_{k=1,2,\ldots}\left\{ \lambda_k \varrho^p + \mathbf{E}_{P_n}[\varphi_{\lambda_k,f}] + \frac{24}{\sqrt{n}} \mathfrak{C}(\mathcal{F}) + \frac{Mt_k}{\sqrt{n}} \right\} \\
&= \min_{k=1,2,\ldots}\left\{ k\varrho^p + \mathbf{E}_{P_n}[\varphi_{k,f}] + \frac{24}{\sqrt{n}} \mathfrak{C}(\mathcal{F}) + \frac{Mt}{\sqrt{n}} + \frac{M\sqrt{\log k}}{\sqrt{n}} \right\} \\
&\leq \min_{\lambda \geq 0}\left\{ (\lambda + 1)\varrho^p + \mathbf{E}_{P_n}[\varphi_{\lambda,f}] + \frac{24}{\sqrt{n}} \mathfrak{C}(\mathcal{F}) + \frac{Mt}{\sqrt{n}} + \frac{M\sqrt{\log(\lambda + 1)}}{\sqrt{n}} \right\},
\end{aligned}
$$

where the last line holds since, for any $\lambda \geq 0$, there exists $k \in \{1, 2, \ldots\}$ such that $\lambda \leq k \leq \lambda + 1$, and $\varphi_{\lambda_1,f} \leq \varphi_{\lambda_2,f}$ holds whenever $\lambda_1 \geq \lambda_2$ (from the definition of $\varphi_{\lambda,f}$).

For the other direction, notice that

$$R_{\varrho,p}(P_n, f) \leq \min_{\lambda \geq 0}\left\{ \lambda \varrho^p + \mathbf{E}_P[\varphi_{\lambda,f}] + \sup_{f \in \mathcal{F}}(\mathbf{E}_{P^n}[\varphi_{\lambda,f}] - \mathbf{E}_P[\varphi_{\lambda,f}]) \right\},$$

where the random variable $\sup_{f \in \mathcal{F}}(\mathbf{E}_{P^n}[\varphi_{\lambda,f}] - \mathbf{E}_P[\varphi_{\lambda,f}])$ can be analyzed in the same way as above. This leads to

$$\mathbf{P}\left(\exists f \in \mathcal{F} \,:\, R_{\varrho,p}(P_n, f) > \min_{\lambda \geq 0}\left\{(\lambda+1)\varrho^p + \mathbf{E}_P\left[\varphi_{\lambda,f}(Z)\right] + \frac{M\sqrt{\log(\lambda+1)}}{\sqrt{n}}\right\}\right.$$
$$\left. + \frac{24\mathfrak{C}(\mathcal{F})}{\sqrt{n}} + \frac{Mt}{\sqrt{n}}\right) \leq 2\exp(-2t^2),$$

for any $t > 0$.

## C.2  Proof of Lemma 1

First note that we have

$$\tilde{\lambda} \cdot \varrho^p \leq \tilde{\lambda} \cdot \varrho^p + \mathbf{E}_Q\left[\sup_{z' \in \mathcal{Z}}\{\tilde{f}(z') - \tilde{f}(Z) - \tilde{\lambda} \cdot d_{\mathcal{Z}}^p(Z, z')\}\right],$$

as the left-hand side corresponds to the choice $z' = Z$. Now, by the optimality of $\tilde{\lambda}$ with respect to $\tilde{f}$, the right-hand side can be further upper-bounded as follows for any $\lambda \geq 0$:

$$\leq \lambda \cdot \varrho^p + \mathbf{E}_Q\left[\sup_{z' \in \mathcal{Z}}\{\tilde{f}(z') - \tilde{f}(Z) - \lambda \cdot d_{\mathcal{Z}}^p(Z, z')\}\right]$$
$$\leq \lambda \cdot \varrho^p + \mathbf{E}_Q\left[\sup_{z' \in \mathcal{Z}}\{L \cdot d_{\mathcal{Z}}(Z, z') - \lambda \cdot d_{\mathcal{Z}}^p(Z, z')\}\right]$$
$$\leq \lambda \cdot \varrho^p + \sup_{t \geq 0}\{L \cdot t - \lambda \cdot t^p\},$$

where for the second line we used the Lipschitz property (Assumption 3) and the third line holds by parametrizing $t = d_{\mathcal{Z}}(Z, z')$. If $p = 1$, we can simply take $\lambda = L$ to get the inequality

$$\tilde{\lambda} \cdot \varrho \leq L \cdot \varrho + \sup_{t \geq 0}\{L \cdot t - L \cdot t\} = L \cdot \varrho,$$

which gives $\tilde{\lambda} \leq L$. If $p > 1$, we can use the optimal value of $t = (L/p\lambda)^{1/(p-1)}$ to get

$$\leq \lambda \cdot \varrho^p + L^{\frac{p}{p-1}} p^{-\frac{p}{p-1}}(p-1)\lambda^{-\frac{1}{p-1}}.$$

Minimizing the right-hand side over $\lambda \geq 0$ with the choice of $\lambda = L/p\varrho^{p-1}$, we get

$$\tilde{\lambda} \cdot \varrho^p \leq L\varrho,$$

which yields the stated bound on $\tilde{\lambda}$.

## C.3  Proof of Theorem 2

The proof is same as the proof of Theorem 3 given below, except that we use Lemma 1 instead of Lemma 2. Then, the expected Rademacher complexity of the function class satisfies

$$\mathfrak{R}_n(\Phi) \leq \frac{24}{\sqrt{n}}\mathfrak{C}(\mathcal{F}) + \frac{24L \cdot C_0 \cdot \mathsf{diam}(\mathcal{Z})^p}{\sqrt{n}\varrho^{p-1}}$$

(see Section D), and the result follows.

## C.4  Proof of Lemma 2

Since $\varphi_{\lambda,f} \geq 0$ for all $\lambda, f$, we arrive at

$$\tilde{\lambda} \leq \frac{R_{\varrho,p}^*(Q, \mathcal{F})}{\varrho^p}. \tag{C.1}$$

We proceed to upper-bound the local minimax risk $R^*_{\varrho,p}(Q,\mathcal{F})$:

$$R^*_{\varrho,p}(Q,\mathcal{F}) = \inf_{f\in\mathcal{F}}\min_{\lambda\geq 0}\left\{\lambda\varrho^p + \int_{\mathcal{Z}}\sup_{z'\in\mathcal{Z}}\left[f(z') - \lambda d^p_{\mathcal{Z}}(z,z')\right]Q(\mathrm{d}z')\right\}$$

$$\leq \min_{\lambda\geq 0}\left\{\lambda\varrho^p + \int_{\mathcal{Z}}\sup_{z'\in\mathcal{Z}}\left[f_0(z') - \lambda d^p_{\mathcal{Z}}(z,z')\right]Q(\mathrm{d}z')\right\}$$

$$\leq \min_{\lambda\geq 0}\left\{\lambda\varrho^p + \int_{\mathcal{Z}}\sup_{z'\in\mathcal{Z}}\left[C_0 d^p_{\mathcal{Z}}(z',z_0) - \lambda d^p_{\mathcal{Z}}(z,z')\right]Q(\mathrm{d}z')\right\}.$$

For $\lambda\geq C_0 2^{p-1}$, the integrand can be upper-bounded as follows:

$$\sup_{z'\in\mathcal{Z}}\left[C_0 d^p_{\mathcal{Z}}(z',z_0) - \lambda d^p_{\mathcal{Z}}(z,z')\right] \leq \sup_{z'\in\mathcal{Z}}\left[C_0 2^{p-1}d^p_{\mathcal{Z}}(z,z_0) + (C_0 2^{p-1}-\lambda)d^p_{\mathcal{Z}}(z,z')\right]$$

$$\leq C_0 2^{p-1}d^p_{\mathcal{Z}}(z,z_0).$$

Therefore,

$$R^*_{\varrho,p}(Q,\mathcal{F}) \leq \min_{\lambda\geq C_0 2^{p-1}}\left\{\lambda\varrho^p + C_0 2^{p-1}\int_{\mathcal{Z}}d^p_{\mathcal{Z}}(z,z_0)Q(\mathrm{d}z)\right\}$$

$$\leq C_0 2^{p-1}\left(\varrho^p + (\mathsf{diam}(\mathcal{Z}))^p\right).$$

Substituting this estimate into (C.1), we obtain what we want.

## C.5 Proof of Theorem 3

Let $f^*\in\mathcal{F}$ be any achiever of the local minimax risk $R^*_{\varrho,p}(P,\mathcal{F})$. We start by decomposing the excess risk:

$$R_{\varrho,p}(P,\widehat{f}) - R^*_{\varrho,p}(P,\mathcal{F}) = R_{\varrho,p}(P,\widehat{f}) - R_{\varrho,p}(P,f^*)$$

$$\leq R_{\varrho,p}(P,\widehat{f}) - R_{\varrho,p}(P_n,\widehat{f}) + R_{\varrho,p}(P_n,f^*) - R_{\varrho,p}(P,f^*),$$

where the last step follows from the definition of $\widehat{f}$. Define

$$\widehat{\lambda} := \arg\min_{\lambda\geq 0}\left\{\lambda\varrho^p + \mathbf{E}_{P_n}[\varphi_{\lambda,\widehat{f}}(Z)]\right\}, \qquad \lambda^* := \arg\min_{\lambda\geq 0}\left\{\lambda\varrho^p + \mathbf{E}_P[\varphi_{\lambda,f^*}(Z)]\right\}.$$

Then, using Proposition 4, we can write

$$R_{\varrho,p}(P,\widehat{f}) - R_{\varrho,p}(P_n,\widehat{f}) = \min_{\lambda\geq 0}\left\{\lambda\varrho^p + \int_{\mathcal{Z}}\varphi_{\lambda,\widehat{f}}(z)P(\mathrm{d}z)\right\} - \left(\widehat{\lambda}\varrho^p + \int_{\mathcal{Z}}\varphi_{\widehat{\lambda},\widehat{f}}(z)P_n(\mathrm{d}z)\right)$$

$$\leq \int_{\mathcal{Z}}\varphi_{\widehat{\lambda},\widehat{f}}(z)(P-P_n)(\mathrm{d}z)$$

and, following similar logic,

$$R_{\varrho,p}(P_n,f^*) - R_{\varrho,p}(P,f^*) \leq \int_{\mathcal{Z}}\varphi_{\lambda^*,f^*}(z)(P_n-P)(\mathrm{d}z). \tag{C.2}$$

By Lemma 2, $\widehat{\lambda}\in\Lambda := [0, C_0 2^{p-1}(1 + (\mathsf{diam}(\mathcal{Z})/\varrho)^p)]$. Hence, defining the function class $\Phi := \{\varphi_{\lambda,f} : \lambda\in\Lambda, f\in\mathcal{F}\}$, we have

$$R_{\varrho,p}(P,\widehat{f}) - R_{\varrho,p}(P_n,\widehat{f}) \leq \sup_{\varphi\in\Phi}\left[\int_{\mathcal{Z}}\varphi\,\mathrm{d}(P-P_n)\right]. \tag{C.3}$$

Since all $f\in\mathcal{F}$ take values in $[0,M]$, the same holds for all $\varphi\in\Phi$. Therefore, by a standard symmetrization argument,

$$R_{\varrho,p}(P,\widehat{f}) - R_{\varrho,p}(P_n,\widehat{f}) \leq 2\,\mathfrak{R}_n(\Phi) + M\sqrt{\frac{2\log(2/\delta)}{n}} \tag{C.4}$$

with probability at least $1 - \delta/2$, where

$$\mathfrak{R}_n(\Phi) := \mathbf{E}\left[\sup_{\varphi \in \Phi} \frac{1}{n} \sum_{i=1}^{n} \varepsilon_i \varphi(Z_i)\right]$$

is the expected Rademacher average of $\Phi$, with i.i.d. Rademacher random variables $\varepsilon_1, \ldots, \varepsilon_n$ independent of $Z_1, \ldots, Z_n$. Moreover, from (C.2) and from Hoeffding's inequality it follows that

$$R_{\varrho,p}(P_n, f^*) - R_{\varrho,p}(P, f^*) \leq M\sqrt{\frac{\log(2/\delta)}{2n}} \tag{C.5}$$

with probability at least $1-\delta/2$. Combining (C.4) and (C.5), and applying Lemma 5 from Appendix D, we obtain the theorem.

## C.6 Proof of Corollary 1

We first verify the regularity assumptions. Assumption 1 is evidently satisfied since $\operatorname{diam}(\mathcal{Z}) = \sqrt{\operatorname{diam}(\mathcal{X})^2 + \operatorname{diam}(\mathcal{Y})^2} \leq 2\sqrt{r_0^2 + B^2}$. Each $f \in \mathcal{F}$ is continuous, and Assumption 2 holds with $M = (\|s\|_\infty + B)^2$. To verify Assumption 3, we proceed as

$$
\begin{aligned}
|f(x, y) - f(x', y')| &= \left|(y - s(f_0^T x))^2 - (y' - s(f_0^T x'))^2\right| \\
&\leq \left|y + y' - s(f_0^T x) - s(f_0^T x')\right| \cdot \left|y - y' + s(f_0^T x') - s(f_0^T x)\right| \\
&\leq (2B + 2\|s\|_\infty) \cdot \left(|y - y'| + \left|s(f_0^T x') - s(f_0^T x)\right|\right) \\
&\leq (2B + 2\|s\|_\infty) \cdot (1 + \|s'\|_\infty) \left(|y - y'| + \|x' - x\|\right) \\
&\leq 2\sqrt{2}(B + \|s\|_\infty) \cdot (1 + \|s'\|_\infty) \sqrt{|y - y'|^2 + \|x - x'\|^2},
\end{aligned}
$$

where the last line follows from Jensen's inequality. Hence, Assumption 3 holds with $L = 2\sqrt{2}(B + \|s\|_\infty)(1 + \|s'\|_\infty)$.

To evaluate the Dudley entropy integral in (3), we need to estimate the covering numbers $\mathcal{N}(\mathcal{F}, \|\cdot\|_\infty, \cdot)$. First observe that, for any two $f, g \in \mathcal{F}$ corresponding to $f_0, g_0 \in \mathbb{R}^d$, we have

$$
\begin{aligned}
\sup_{x \in \mathcal{X}} \sup_{y \in \mathcal{Y}} |f(x, y) - g(x, y)| &= \sup_{x \in \mathcal{X}} \sup_{y \in \mathcal{Y}} \left|\left(y - s(f_0^T x)\right)^2 - \left(y - s(g_0^T x)\right)^2\right| \\
&\leq 2B \sup_{x \in \mathcal{X}} |s(f_0^T x) - s(g_0^T x)| + \sup_{x \in \mathcal{X}} |s^2(f_0^T x) - s^2(g_0^T x)| \\
&\leq \underbrace{2r_0 (B + \|s\|_\infty) \|s'\|_\infty}_{:=D} \|f_0 - g_0\|_2.
\end{aligned}
$$

Since $f_0, g_0$ belong to the unit ball in $\mathbb{R}^d$,

$$\mathcal{N}(\mathcal{F}, \|\cdot\|_\infty, u) \leq \left(\frac{3D}{u}\right)^d$$

for $0 < u < D$, and $\mathcal{N}(\mathcal{F}, \|\cdot\|_\infty, u/2) = 1$ for $u \geq 2D$, which gives

$$
\begin{aligned}
\int_0^\infty \sqrt{\log \mathcal{N}(\mathcal{F}, \|\cdot\|_\infty, u)}\,\mathrm{d}u &\leq \int_0^D \sqrt{d \log\left(3D/u\right)}\,\mathrm{d}u \\
&= 3D\sqrt{d} \int_0^{1/3} \sqrt{\log\left(1/u\right)}\,\mathrm{d}u \\
&\leq 3D\sqrt{d}/2.
\end{aligned}
$$

Substituting this into the bound (10), we get the desired estimate.

## C.7 Proof of Corollary 2

We will denote by $\langle \cdot, \cdot \rangle_K$ the inner product in $\mathcal{H}_K$, and by $\|\cdot\|_K$ the induced norm.

For completeness, we state the covering number estimates by Cucker and Zhou [7, Thm 5.1].

**Proposition 6.** *For compact $\mathcal{X} \subset \mathbb{R}^d$, the following holds for all $u \in (0, r/2]$.*

$$\log \mathcal{N}(I_K(\mathcal{B}_r), \|\cdot\|_{\mathcal{X}}, u) \le d \left( 32 + \frac{640 d (\mathsf{diam}(\mathcal{X}))^2}{\sigma^2} \right)^{d+1} \left( \log \frac{r}{u} \right)^{d+1}.$$

We would also need the following technical lemma.

**Lemma 4.** *For any $f, g \in \mathcal{F}$ induced by $f_0, g_0 \in I_K(\mathcal{B}_r)$ (respectively), we have:*

$$\|f\|_\infty \le 2(r^2 + B^2)$$
$$\|f - g\|_\infty \le 2(r + B)\|f_0 - g_0\|_{\mathcal{X}}.$$

*Proof.* First note that $\sqrt{K(x,x)} = 1$ holds for any $x \in \mathcal{X}$ by the definition of Gaussian kernel. This leads immediately to the first claim: for any $x \in \mathcal{X}, y \in [-B, B]$,

$$(f_0(x) - y)^2 \le 2f_0^2(x) + 2y^2 \le 2(\langle f_0, K_x \rangle_K)^2 + 2B^2 \le 2(\langle f_0, f_0 \rangle_K) + 2B^2,$$

where the first inequality is by Jensen's inequality, the second is due to the reproducing kernel property of $K$, and the third is Cauchy-Schwarz inequality in $\mathcal{H}_K$ ($K_x$ denotes the kernel centered at $x$, i.e. $x' \mapsto K(x, x')$). The second claim can be established similarly: for any $x \in \mathcal{X}, y \in [-B, B]$,

$$\begin{aligned}
\left| (f_0(x) - y)^2 - (g_0(x) - y)^2 \right| &= \left| f_0(x) + g_0(x) - 2y \right| \left| f_0(x) - g_0(x) \right| \\
&\le \left( 2 \sup_{h_0 \in I_K(\mathcal{B}_r)} |h_0(x)| + 2|y| \right) \left| f_0(x) - g_0(x) \right| \\
&\le 2(r + B)\|f_0 - g_0\|_{\mathcal{X}},
\end{aligned}$$

where the last inequality is due to Cauchy-Schwarz inequality again. $\square$

Before proceeding, we first observe that the Gaussian kernel is $(\sqrt{2}/\sigma)$-Lipschitz, i.e. $\|K_x - K_{x'}\|_K \le \sqrt{2}/\sigma \cdot \|x - x'\|_2$. Indeed, we can proceed as

$$\begin{aligned}
\|K_x - K_{x'}\|_K^2 &= \langle K_x - K_{x'}, K_x - K_{x'} \rangle_K \\
&= 2 - 2K(x, x') \\
&= 2 - 2\exp\left( -\frac{\|x - x'\|_2^2}{\sigma^2} \right) \\
&\le \frac{2}{\sigma^2} \|x - x'\|_2^2,
\end{aligned}$$

where we used the fact that the function $u \mapsto 2u/\sigma^2 - 2 + 2e^{-u/\sigma^2}$ is nonnegative for $u \ge 0$.

We now check the validity of Assumptions 1–3. Assumption 1 holds as $\mathsf{diam}(\mathcal{Z}) = \sqrt{\mathsf{diam}(\mathcal{X})^2 + \mathsf{diam}(\mathcal{Y})^2} \le 2\sqrt{r_0^2 + B^2}$. The functions in $\mathcal{F}$ are continuous, and Assumption 2 holds with $M = 2(r^2 + B^2)$ by virtue of the first estimate of Lemma 4. To verify Assumption 3, we proceed as

$$\begin{aligned}
|f(x, y) - f(x', y')| &= \left| (y - f_0(x))^2 - (y' - f_0(x'))^2 \right| \\
&\le |y + y' - f_0(x) - f_0(x')| \cdot |y - y' + f_0(x') - f_0(x)| \\
&\le 2(r + B) \cdot (|y - y'| + |\langle f_0, K_{x'} - K_x \rangle_K|) \\
&\le 2(r + B) \cdot (|y - y'| + r \cdot \|K_{x'} - K_x\|_K) \\
&\le 2(r + B) \cdot \left( |y - y'| + \frac{r\sqrt{2}}{\sigma} \|x - x'\|_2 \right) \\
&\le 2(r + B) \cdot \left( 1 + r\sqrt{2}/\sigma \right) (|y - y'| + \|x - x'\|_2) \\
&\le 2\sqrt{2}(r + B) \cdot \left( 1 + r\sqrt{2}/\sigma \right) \sqrt{|y - y'|^2 + \|x - x'\|_2^2},
\end{aligned}$$

where the fourth inequality holds by the Lipschitz continuity of the Gaussian reproducing kernel, and the last inequality is Jensen's inequality. Hence, Assumption 3 holds with $L = 2\sqrt{2}(r + B) \cdot \left( 1 + r\sqrt{2}/\sigma \right)$.

Now we proceed to upper-bound the Dudley entropy integral for $\mathcal{F}$:

$$\int_0^\infty \sqrt{\log \mathcal{N}\left(\mathcal{F}, \|\cdot\|_\infty, u\right)} du \le \int_0^{2(r^2+Br)} \sqrt{\log \mathcal{N}\left(I_K(\mathcal{B}_r), \|\cdot\|_x, \frac{u}{2(r+B)}\right)} du$$

$$\le \underbrace{\int_0^{r^2+Br} \sqrt{\log \mathcal{N}\left(I_K(\mathcal{B}_r), \|\cdot\|_x, \frac{u}{2(r+B)}\right)} du}_{:=T_1}$$

$$+ \underbrace{\int_{r^2+Br}^{2(r^2+Br)} \sqrt{\log \mathcal{N}\left(I_K(\mathcal{B}_r), \|\cdot\|_x, \frac{r}{2}\right)} du}_{:=T_2}$$

where we used the second claim of Lemma 4 for the first inequality and the monotonicity of covering numbers for the second inequality. Plugging in the estimate from Proposition 6, we get

$$T_1 \le 2\sqrt{d}\left(32 + \frac{2560dr_0^2}{\sigma^2}\right)^{\frac{d+1}{2}} (r^2+Br)\Gamma\left(\frac{d+3}{2}, \log 2\right)$$

$$T_2 \le \sqrt{d}\left(32 + \frac{2560dr_0^2}{\sigma^2}\right)^{\frac{d+1}{2}} (r^2+Br)(\log 2)^{\frac{d+1}{2}},$$

and hence $T_1 + T_2 \le \frac{C_1}{48}(r^2+Br)$, where the constant $C_1$ is

$$C_1 = 48\sqrt{d}\left(2\Gamma\left(\frac{d+3}{2}, \log 2\right) + (\log 2)^{\frac{d+1}{2}}\right)\left(32 + \frac{2560dr_0^2}{\sigma^2}\right)^{\frac{d+1}{2}},$$

## D  Rademacher complexity of $\Phi$

**Lemma 5.** *The expected Rademacher complexity of the function class $\Phi$ satisfies*

$$\mathfrak{R}_n(\Phi) \le \frac{24}{\sqrt{n}}\mathfrak{C}(\mathcal{F}) + \frac{12C_0(2\,\mathsf{diam}(\mathcal{Z}))^p}{\sqrt{n}}\left(1 + \left(\frac{\mathsf{diam}(\mathcal{Z})}{\varrho}\right)^p\right).$$

*Proof of Lemma 5.* Define the $\Phi$-indexed process $X = (X_\varphi)_{\varphi \in \Phi}$ via

$$X_\varphi := \frac{1}{\sqrt{n}}\sum_{i=1}^n \varepsilon_i \varphi(Z_i),$$

which is clearly zero-mean: $\mathbf{E}[X_\varphi] = 0$ for all $\varphi \in \Phi$. To upper-bound the Rademacher average $\mathfrak{R}_n(\Phi)$, we first show that $X$ is a subgaussian process with respect to a suitable pseudometric. For $\varphi = \varphi_{\lambda,f}$ and $\varphi' = \varphi_{\lambda',f'}$, define

$$d_\Phi(\varphi, \varphi') := \|f - f'\|_\infty + (\mathsf{diam}(\mathcal{Z}))^p|\lambda - \lambda'|,$$

and it is not hard to show that $\|\varphi - \varphi'\|_\infty \le d_\Phi(\varphi, \varphi')$. Then, for any $t \in \mathbb{R}$, using Hoeffding's lemma and the fact that $(\varepsilon_i, Z_i)$ are i.i.d., we arrive at

$$\mathbf{E}\left[\exp(t(X_\varphi - X_{\varphi'}))\right] = \mathbf{E}\left[\exp\left(\frac{t}{\sqrt{n}}\sum_{i=1}^n \varepsilon_i(\varphi(Z_i) - \varphi'(Z_i))\right)\right]$$

$$= \left(\mathbf{E}\left[\exp\left(\frac{t}{\sqrt{n}}\varepsilon_1(\varphi(Z_1) - \varphi'(Z_1))\right)\right]\right)^n$$

$$\le \exp\left(\frac{t^2 d_\Phi^2(\varphi, \varphi')}{2}\right).$$

Hence, $X$ is subgaussian with respect to $d_\Phi$, and therefore the Rademacher average $\mathfrak{R}_n(\Phi)$ can be upper-bounded by the Dudley entropy integral [19]:

$$\mathfrak{R}_n(\Phi) \le \frac{12}{\sqrt{n}}\int_0^\infty \sqrt{\log \mathcal{N}(\Phi, d_\Phi, u)} du,$$

where $\mathcal{N}(\Phi, d_\Phi, \cdot)$ are the covering numbers of $(\Phi, d_\Phi)$. From the definition of $d_\Phi$, it follows that

$$\mathcal{N}(\Phi, d_\Phi, u) \leq \mathcal{N}(\mathcal{F}, \|\cdot\|_\infty, u/2) \cdot \mathcal{N}(\Lambda, |\cdot|, u/2(\mathrm{diam}(\mathcal{Z}))^p),$$

and therefore

$$\mathfrak{R}_n(\Phi) \leq \frac{12}{\sqrt{n}} \left( \int_0^\infty \sqrt{\log \mathcal{N}(\mathcal{F}, \|\cdot\|_\infty, u/2)} \mathrm{d}u + \int_0^\infty \sqrt{\log \mathcal{N}(\Lambda, |\cdot|, u/2(\mathrm{diam}(\mathcal{Z}))^p)} \mathrm{d}u \right).$$

Since $\Lambda$ is a compact interval, it is straightforward to upper-bound the second integral:

$$\int_0^\infty \sqrt{\log \mathcal{N}(\Lambda, |\cdot|, u/2(\mathrm{diam}(\mathcal{Z}))^p)} \mathrm{d}u \leq 2|\Lambda|(\mathrm{diam}(\mathcal{Z}))^p \int_0^{1/2} \sqrt{\log(1/u)} \mathrm{d}u$$
$$= 2c|\Lambda|(\mathrm{diam}(\mathcal{Z}))^p,$$

where $|\Lambda| = C_0 2^{p-1}(1 + (\mathrm{diam}(\mathcal{Z})/\varrho)^p)$ is the length of the interval $\Lambda$ and the constant $c = \frac{1}{2}\left(\sqrt{\log 2} + \sqrt{\pi} \cdot \mathrm{erfc}(\sqrt{\log 2})\right) < 1$. Consequently,

$$\mathfrak{R}_n(\Phi) \leq \frac{12}{\sqrt{n}} \left( \int_0^\infty \sqrt{\log \mathcal{N}(\mathcal{F}, \|\cdot\|_\infty, u/2)} \mathrm{d}u + 2|\Lambda|(\mathrm{diam}(\mathcal{Z}))^p \right)$$
$$\leq \frac{24}{\sqrt{n}} \mathfrak{C}(\mathcal{F}) + \frac{12C_0(2\,\mathrm{diam}(\mathcal{Z}))^p}{\sqrt{n}} \left( 1 + \left( \frac{\mathrm{diam}(\mathcal{Z})}{\varrho} \right)^p \right).$$

$\square$

# E    Proofs for Section 4

## E.1    Proof of Lemma 3

First we prove that $W_p(P, Q) \leq W_p(\mu, \nu)$. Define the mapping $\tilde{T} : \mathcal{Z} \to \mathcal{Z}$ by $\tilde{T} := T \otimes \mathrm{id}_\mathcal{Y}$, i.e., $\tilde{T}(z) = \tilde{T}(x, y) = (T(x), y)$, and let $\tilde{Q} = \tilde{T}_\# P$, the pushforward of $P$ by $\tilde{T}$. We claim that $\tilde{Q} \equiv Q$. Indeed, for any measurable sets $A \subseteq \mathcal{X}$ and $B \subseteq \mathcal{Y}$,

$$\tilde{Q}(A \times B) = \tilde{T}_\# P(A \times B)$$
$$= P(T^{-1}(A) \times B)$$
$$= \int_{T^{-1}(A)} \mu(\mathrm{d}x) P_{Y|X}(B|x)$$
$$= \int_A T_\# \mu(\mathrm{d}x) P_{Y|X}(B|T(x))$$
$$= \int_A \nu(\mathrm{d}x) Q_{Y|X}(B|x),$$

where we have used the relation (13) and the invertibility of $T$. Thus,

$$W_p^p(P, Q) \leq \mathbf{E}_P[d_\mathcal{Z}^p(Z, \tilde{T}(Z)))] = \mathbf{E}_P[d_\mathcal{X}^p(X, T(X))] = W_p^p(\mu, \nu).$$

For the reverse inequality, let $M \in \mathcal{P}(\mathcal{Z} \times \mathcal{Z})$ be the optimal coupling of $P$ and $Q$. Then, for $Z = (X, Y)$ and $Z' = (X', Y')$ with $(Z, Z') \sim M$, the marginal $M_{XX'}$ is evidently a coupling of the marginals $\mu$ and $\nu$, and therefore

$$W_p^p(P, Q) = \mathbf{E}_M[d_\mathcal{Z}^p(Z, Z')]$$
$$= \mathbf{E}_M[d_\mathcal{X}^p(X, X')] + \mathbf{E}_M[d_\mathcal{Y}^p(Y, Y')]$$
$$\geq \mathbf{E}_M[d_\mathcal{X}^p(X, X')]$$
$$\geq W_p^p(\mu, \nu).$$

## E.2 Proof of Theorem 4

For simplicity, we assume that there exists a hypothesis $f^* \in \mathcal{F}$ that achieves $R^*(Q, \mathcal{F})$. Then, for any $\varrho > 0$ such that $W_p(P, Q) \leq \varrho$, Proposition 1 implies that

$$R(Q, \widehat{f}) - R(Q, f^*) \leq R_{\varrho,p}(P, \widehat{f}) - R_{\varrho,p}(P, f^*) + 2L\varrho$$
$$\leq R_{\varrho,p}(P, \widehat{f}) - R^*_{\varrho,p}(P, \mathcal{F}) + 2L\varrho.$$

From Theorem 2, we know that

$$R_{\varrho,p}(P, \widehat{f}) - R^*_{\varrho,p}(P, \mathcal{F}) \leq \frac{48\mathfrak{C}(\mathcal{F})}{\sqrt{n}} + \frac{48L\mathsf{diam}^p(\mathcal{Z})}{\sqrt{n}\varrho^{p-1}} + \frac{3M\sqrt{\log(4/\delta)}}{\sqrt{2n}}$$

holds with probability at least $1 - \delta/2$. Thus, it remains to find the right $\varrho$, such that that $W_p(P, Q) \leq \varrho$ holds with high probability. From Proposition 5, we see that each of the following two statements holds with probability at least $1 - \delta/4$:

$$W_p(\mu_n, \mu) \leq \left( \frac{\log(4C_a/\delta)}{C_b n} \right)^{p/d}, \qquad W_p(\nu_m, \nu) \leq \left( \frac{\log(4C_a/\delta)}{C_b m} \right)^{p/d}.$$

Since $W_p(P, Q) = W_p(\mu, \nu)$ by Lemma 3, we see that $W_p(P, Q) \leq \widehat{\varrho}(\delta)$ with probability at least $1 - \delta/2$, where $\widehat{\varrho}(\delta)$ is given by Eq. (17). The claim of the theorem follows from the union bound.

## Footnotes

[2]Note that we are assuming that $\varrho \leq (1 - \max Z_i) \cdot n^{-1/p}$; otherwise, the local minimax risk of $f_1$ can be smaller, which leads to even higher probability of choosing nonrobust hypothesis by local minimax ERM