[Reviews · NeurIPS 2018]

Reviewer 1



Wasserstein distance is used to measure the difference between two probability measures and provides efficient tools for solving many problems in machine learning. In this paper, the authors propose a novel minimax framework for statistical learning with ambiguity sets given by balls in Wasserstein space and offer an interesting point of view for the domain adaptation problem [6]. The relation between the proposed method and the ordinary empirical risk minimization is discussed and the generalization bounds are derived in terms of covering number. The result is interesting and gives some insight in distributionally robust learning.

Reviewer 2



The paper investigates a minimax framework for statistical learning where the goal is to minimize the worst-case population risk over a family of distributions that are within a prescribed Wasserstein distance from the unknown data-generating distribution. The authors develop data-dependent generalization bound and data-independent excess risk bounds (using smoothness assumptions) in the setting where the classical empirical risk minimization (ERM) algorithm is replaced by a robust procedure that minimizes the worst-case empirical risk with respect to distributions contained in a Wasserstein ball centered around the data-generating empirical distribution. The statistical minimax framework investigated by the authors resembles in spirit the one introduced in [9], where the ambiguity set is defined via moment constraints instead of the Wasserstein distance. The paper is well-written, with accurate references to previous literature and an extensive use of remarks to guide the development of the theory. The contributions are clearly emphasized, and the math is solid. The need for a new statistical framework that replaces the goal to minimize the population risk with a worst-case population risk is well motivated in Section 2.1, and the example on domain adaptation with optimal transport motivates the use of the Wasserstein distance to define the ambiguity set. The only remark I have is that there is a misalignment in terms of the theory developed by the authors, where the goal is to minimize the worst-case population risk, and the example provided on optimal transport, where the goal is to derive improved excess risk bounds with respect to the ordinary (i.e., not worst-case) population risk. I would encourage the authors to comment more explicitly in the main text on how the theory that they have developed in the previous sections (see Appendix E) is used to prove the ordinary excess bounds. In line 234 the authors write that the risk inequalities of Section 2 are used to prove the results of Theorem 4, which is not the case.

Reviewer 3



This paper studies the problem of minimizing the worst-case risk over a Wasserstein ball around the underlying distribution. General risk upper bounds are provided for the procedure which minimizes the error over the Wasserstein ball around empirical distribution, which directly leads to excess risk results for Lipschitz classes. The paper also applies this result for specific model classes such as simple neural network class, RKHS, etc. And the results are also applied to domain adaptation. The key idea in the proof of main theorems (Thm 1~3) is based on the strong duality for distributional robust stochastic optimization. Once we write it in dual form, standard Dudley chaining bounds lead to the desired result. This part is very simple but there're something smart happening here: we only need to consider the distribution class within a Wasserstein ball around empirical distribution, and to compare with best local minimax risk within this class, we don't need to pay for the constant depending on $\rho$ which does not go to zero with n. However, since the proof of this abstract result is pretty standard, and the key ingredient (duality) is directly from previous works, I doubt if the contribution is strong enough for NIPS. The author also provides examples, but they are simply plugging existing covering number results into the bound. It would be much better if the authors can illustrate through some examples about the surprising properties of distributional robust setup. The domain adaptation part seems even less satisfactory. An important point in previous sections is that, the risk does not depend on Wasserstein distance between empirical measure and the underlying measure, which has exponential dependence on dimension. But the domain adaptation part gets back this dependence, losing the merits of previous bounds. Detailed comments: In Theorem 2, the excess risk blows up when $\rho$ goes to zero, for $p>1$. This contradicts the intuition that less adversarial noise makes the problem easier. And the choice of $\lambda$ and the resulting risk cannot be always optimal.